# Calcium modeling of spine apparatus-containing human dendritic spines demonstrates an "all-or-nothing" communication switch between the spine head and dendrite

**James Rosado**[1⊙], **Viet Duc Bui**[2⊙], **Carola A. Haas**[3,4,5,6], **Jürgen Beck**[3,6], **Gillian Queisser**[1‡*], **Andreas Vlachos**[2,4,5,6‡*]

**1** Department of Mathematics, Temple University, Philadelphia, Pennsylvania, United States of America, **2** Department of Neuroanatomy, Institute of Anatomy and Cell Biology, Faculty of Medicine, University of Freiburg, Freiburg, Germany, **3** Department of Neurosurgery, Medical Center-University of Freiburg, Faculty of Medicine, University of Freiburg, Freiburg, Germany, **4** Bernstein Center Freiburg, University of Freiburg, Freiburg, Germany, **5** Center Brain Links Brain Tools, University of Freiburg, Freiburg, Germany, **6** Center for Basics in Neuromodulation (NeuroModulBasics), Faculty of Medicine, University of Freiburg, Freiburg, Germany

⊙ These authors contributed equally to this work.
‡These authors are joint senior authors on this work.
* gillian.queisser@temple.edu (GQ); andreas.vlachos@anat.uni-freiburg.de (AV)

**Data Availability Statement:** Code used for running numerical experiments, generating data,

## Abstract

Dendritic spines are highly dynamic neuronal compartments that control the synaptic transmission between neurons. Spines form ultrastructural units, coupling synaptic contact sites to the dendritic shaft and often harbor a spine apparatus organelle, composed of smooth endoplasmic reticulum, which is responsible for calcium sequestration and release into the spine head and neck. The spine apparatus has recently been linked to synaptic plasticity in adult human cortical neurons. While the morphological heterogeneity of spines and their intracellular organization has been extensively demonstrated in animal models, the influence of spine apparatus organelles on critical signaling pathways, such as calcium-mediated dynamics, is less well known in human dendritic spines. In this study we used serial transmission electron microscopy to anatomically reconstruct nine human cortical spines in detail as a basis for modeling and simulation of the calcium dynamics between spine and dendrite. The anatomical study of reconstructed human dendritic spines revealed that the size of the postsynaptic density correlates with spine head volume and that the spine apparatus volume is proportional to the spine volume. Using a newly developed simulation pipeline, we have linked these findings to spine-to-dendrite calcium communication. While the absence of a spine apparatus, or the presence of a purely passive spine apparatus did not enable any of the reconstructed spines to relay a calcium signal to the dendritic shaft, the calcium-induced calcium release from this intracellular organelle allowed for finely tuned "all-or-nothing" spine-to-dendrite calcium coupling; controlled by spine morphology, neck plasticity, and ryanodine receptors. Our results suggest

and plotting is available on the GitHub repository at https://github.com/NeuroBox3D/calciumDynamics_app/tree/master/spine.

**Funding:** This work was supported by the NIH grant number R01MH118930 to GQ and 1R01NS109498 to AV. This work was also supported by the Federal Ministry of Education and Research, Germany grant BMBF 01GQ1804A to AV. The funders had no role in study design, data collection and analysis, decision to publish, or preparation of the manuscript.

**Competing interests:** The authors have declared that no competing interests exist.

that spine apparatus organelles are strategically positioned in the neck of human dendritic spines and demonstrate their potential relevance to the maintenance and regulation of spine-to-dendrite calcium communication.

## Author summary

During the past decade it has become increasingly clear that abnormal synaptic plasticity is a major hallmark of neurological and cognitive disorders. Developing a better understanding of the synaptic plasticity process, which describes the ability of neurons to adapt their contacts in an activity-dependent manner, will lead to improved treatment of many neurological and cognitive disorders. It is known that calcium-dependent events such as synaptic transmission, intracellular calcium release, and calcium wave propagation, are required for many types of synaptic plasticity expression. However, the biological significance of these processes in neurons of the adult human cortex remains unknown. Due to technical limitations and ethical concerns, experimental data addressing this biologically and clinically relevant topic are not available. Therefore, we have implemented a computational model to study the intracellular calcium dynamics in realistic human dendritic spines based on detailed morphological reconstructions. With our model and simulations, we have established the morphological and biological requirements for the propagation of calcium from spines into the dendrites. Our results suggest a critical role for the calcium-storing spine apparatus organelle in regulating calcium homeostasis and propagation in human dendritic spines.

## Introduction

Calcium-based second-messenger signaling pathways mediate fundamental biological processes in the human body [1]. In the brain, synaptically evoked intracellular $Ca^{2+}$ waves regulate the ability of neurons to adjust their synaptic contacts in an activity-dependent manner [2, 3]. This fundamental biological mechanism (i.e., synaptic plasticity), plays a crucial role in complex brain functions, such as internal representation and memory formation [4, 5]. Over the past few decades, cellular and molecular mechanisms of $Ca^{2+}$ dependent synaptic plasticity have been extensively studied in dendritic spines from various model organisms [6, 7].

Dendritic spines are highly dynamic neuronal compartments on which cortical excitatory synapses are typically formed [8–10]. Most spines have a bulbous head and a thin neck that connects the spine head to the shaft of the dendrite. Additionally, they often harbor a dynamic extension of the dendritic smooth endoplasmic reticulum (sER), which can rapidly enter and leave the spine compartment and is capable of changing its position inside the neck and head of spines and plays a critical role in sER intracellular $Ca^{2+}$ storage and in synaptic plasticity [11–14]. The heterogeneous nature of spine morphology and intracellular $Ca^{2+}$ stores has been studied with respect to electrical and biochemical signaling [15–17]. Recent research has focused on the influence of synaptic spine architecture on calcium signaling by performing computational studies on a wide variety of synthetically designed geometries, considering key spine features such as spine neck width/length, spine head volume, and the size of the sER and performing computational studies on a wide variety of such synthetically designed geometries [8, 18–20].

Guided by the existing literature and our own recent work [20–22], we made detailed anatomical reconstructions of human cortical spines based on serial transmission electron microscopy and performed a correlation analysis on a set of geometric features, including the size and position of the postsynaptic density (PSD), spine head volumes, and cross-sectional areas of the spine neck. We then attempted to understand how the distinct three-dimensional ultrastructural organization of human dendritic spines influences intracellular $Ca^{2+}$ dynamics in the spine head and neck, and how these signals propagate into the attached dendritic shaft (i.e., spine-to-dendrite $Ca^{2+}$ communication). Specifically, we wanted to learn more about the role of the spine apparatus (SA) organelle [23–25], an enigmatic cellular organelle composed of stacked sER, found in the majority of cortical dendritic spines in humans [22].

## Results

### Spine $Ca^{2+}$ modeling in reconstructed human dendritic spines containing a spine apparatus organelle

To understand the effect of the SA on $Ca^{2+}$ dynamics in dendritic spines, the three-dimensional architecture of dendritic spines, including intracellular membrane structures, must be carefully considered. Expanding on the results obtained from synthetically generated dendritic spines [20], we reconstructed detailed morphologies of nine dendritic spines, including their SA, and postsynaptic densities from cortical surgical access material using serial transmission electron microscopy (TEM). From the serial TEM image stacks, we then generated realistic three-dimensional geometric models, see Fig 1A and 1B.

Immunogold staining for the actin-binding protein synaptopodin, which is an essential component for the formation of the SA in the rodent brain [26], confirmed that the actin-modulating protein synaptopodin is also a marker for human SA (Fig 1C; c.f., [22]). We then attached the reconstructed SAs to a single standardized dendritic ER tubule in a dendritic compartment of identical dimensions for all spines (Fig 1D). We used the resulting surface triangulation of the plasma membrane, the SA, the PSD, and dendritic ER to construct a volumetric tetrahedral mesh [27] for numerical simulations.

Next, we adapted and integrated existing single-channel models of $Ca^{2+}$ exchangers in the plasma membrane—i.e., plasma membrane $Ca^{2+}$-ATPase (PMCA) and $Na^+/Ca^{2+}$ exchangers (NCX), as well as ryanodine receptors (RyR) on the SA membrane (see schematics in Fig 1E) —into a diffusion-reaction model for cytosolic and endoplasmic $Ca^{2+}$ dynamics [20]. Numerical simulations were performed on computational domains defined by the reconstructed human dendritic spine morphologies, using established numerical methods (details provided in Methods).

### Spine apparatus-containing human dendritic spines showed larger spine head volumes and postsynaptic densities

We recently showed that ∼70% of all dendritic spines on human layer II/III pyramidal neurons contain a synaptopodin cluster (i.e., SA) [22]. By carefully examining EM cross-sections, we obtained evidence that large human dendritic spines contain SA organelles [22], which is consistent with earlier work on synaptopodin clusters, SA, and spine head sizes of neurons in rodent brains [13, 28–30]. However, no three-dimensional analyses of SA-containing human dendritic spines have been performed thus far.

To test whether there were correlations between the reconstructed spine components, we carefully examined critical morphological parameters of the reconstructed SA-containing dendritic spines (Fig 2). Consistent with previous studies in animal models [31, 32], we found a

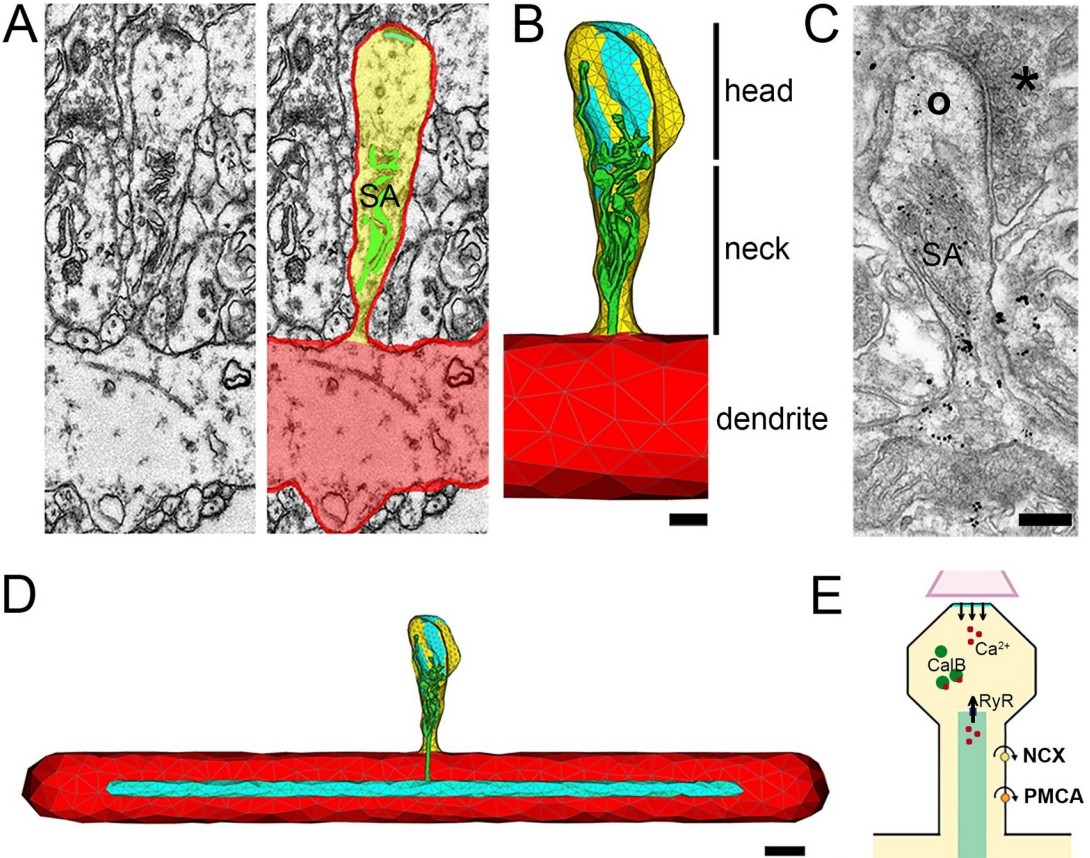

**Fig 1. Dendritic spine reconstruction and Ca²⁺ modeling.** (A) Spine morphologies are based on serial reconstructions of transmission electron micrographs obtained from human cortical tissue. (B) An example of a reconstructed dendritic spine (dendrite, red; spine compartment, yellow) carrying a spine apparatus organelle (SA, green). Upon release of Ca²⁺ at the reconstructed postsynaptic density (cyan), changes in [Ca²⁺] are determined in the head, neck and dendritic region, respectively. Scale bar, 200 nm. (C) Immunogold staining for the actin-binding protein synaptopodin [22], which is a marker and essential component of the SA. Asterisk indicates presynaptic, circle the postsynaptic compartment. Scale bar, 200 nm. (D) In all reconstructed human spines the SA was attached to a single standardized dendritic ER tubule in a dendritic compartment with identical dimensions. Scale bar, 500 nm. (E) The model accounts for Ca²⁺ exchange mechanisms on the plasma membrane (Na⁺/Ca²⁺ exchangers (NCX), plasma membrane Ca²⁺-ATPases (PMCA)) and on the ER (green; ryanodine-receptors (RyR)). Calcium buffering capacity (Calbindin, CalB) is kept constant. Further details on physiological values are provided in the main text and in Table 1. Tables 2 and 3 contain morphological measurements such as surface area. See also S1 and S2 Videos, these videos demonstrate the propagation of calcium to the dendritic region.

positive correlation between PSD sizes and spine volumes in the SA-containing spines reconstructed from human cortical tissue (Fig 2B). This relationship was readily detected when spines were sorted by spine head volume. Moreover, we observed a positive correlation between PSD sizes and cross-sectional areas of spine necks (Fig 2C). This finding raises the possibility that larger spines with a wider neck may allow for a more effective Ca²⁺ spine-to-dendrite communication.

To model the spine-to-dendrite Ca²⁺ communication, we first determined distances between PSDs and the dendritic compartment. Fig 2D shows no significant correlation between spine volumes and PSD-to-dendrite distances, which were defined as the Euclidian distance between the point of the PSD closest to the base of the spine. These results indicated that while SA-containing spines of the human cortex may differ in their volume and spine

**Table 1. Numerical Values for Simulation Parameters.**

| Initial and equilibrium values | | PMCA pumps | |
|---|---|---|---|
| $c_c$ | 50 nM | $I_p$ | $1.7 \times 10^{-23}$ mol $\cdot$ $s^{-1}$ |
| $c_e$ | 250 $\mu$M | $K_p$ | 60 nM |
| $c_o$ | 2 mM | $\rho_p$ | 500 $\mu m^{-2}$ |
| $p$ | 40 nM | **NCX pumps** | |
| $b^{tot}$ | 40 $\mu$M | $I_N$ | $2.5 \times 10^{-21}$ mol $\cdot$ $s^{-1}$ |
| **Diffusion/reaction** | | $K_N$ | 1.8 $\mu$M |
| $D_c$ | 220 $\mu m^2 \cdot s^{-1}$ | $\rho_N$ | 15 $\mu m^{-2}$ |
| $D_p$ | 280 $\mu m^2 \cdot s^{-1}$ | **Leakage** | |
| $D_b$ | 20 $\mu m^2 \cdot s^{-1}$ | $v_{l,e}$ | 38 $nm \cdot s^{-1}$ |
| $\kappa_b^-$ | 19 $s^{-1}$ | $v_{l,p}$ | 4.5 $nm \cdot s^{-1}$ |
| $\kappa_b^+$ | 27 $\mu M^{-1}\ s^{-1}$ | | |
| $\kappa_p$ | 0.11 $s^{-1}$ | **Calcium release** | |
| $p^r$ | 40 nM | $j_c^{rls}$ | $8.6 \times 10^{-18}$ mol $\cdot s^{-1}\ \mu m^{-2}$ |
| **RyR channel** | | $\tau_{rls}$ | 10 $ms$ |
| $k_a^-$ | 28.8 $s^{-1}$ | | |
| $k_a^+$ | 1500 $\mu M^{-4} \cdot s^{-1}$ | | |
| $k_b^-$ | 385.9 $s^{-1}$ | | |
| $k_b^+$ | $1500 \mu M^{-3} \cdot s^{-1}$ | | |
| $k_c^-$ | 0.1 $s^{-1}$ | | |
| $k_c^+$ | 1.75 $s^{-1}$ | | |
| $\rho_r$ | $0.0 - 5.0\ \mu m^{-2}$ | | |
| $c_e^{ref}$ | 0.25 mM | | |
| $I_R^{ref}$ | $3.5 \times 10^{-18}$ mol $\cdot s^{-1}$ | | |

Constants used in simulations are borrowed from [20].

**Table 2. RyR Critical Density, Surface Area Measurements, and Number of RyRs.**

| Spine | RyR Crit. Density | SA Surface Area | ER Surface Area | No. RyRs (SA) | No. RyRs (ER) |
|---|---|---|---|---|---|
| 1 | 2.58 | 0.3775 | 5.2696 | 1 | 14 |
| 2 | 2.99 | 0.3607 | 5.6055 | 1 | 17 |
| 3 | 2.90 | 0.5867 | 5.8341 | 2 | 17 |
| 4 | 2.03 | 1.1270 | 6.3761 | 2 | 13 |
| 5 | 3.13 | 1.0291 | 6.2729 | 3 | 20 |
| 6 | 2.60 | 0.6807 | 5.9312 | 2 | 15 |
| 6A | 2.64 | 0.7092 | 5.9596 | 2 | 16 |
| 6B | 2.66 | 0.7220 | 5.9724 | 2 | 16 |
| 6C | 2.67 | 0.7347 | 5.9852 | 2 | 16 |
| 7 | 1.37 | 2.3209 | 7.5629 | 3 | 10 |
| 8 | 3.50 | 1.7146 | 6.6048 | 6 | 23 |
| 9 | 2.09 | 0.9134 | 5.8054 | 2 | 12 |

Surface area is in units of $\mu m^2$. The ER includes the SA and tube inside dendrite.

**Table 3. Neck Cross-Sectional Areas.**

| Spine | SA Neck | Spine Neck | Cross Section Ratio (%) |
|:---:|:---:|:---:|:---:|
| 3 | 0.0003265 | 0.0074247 | 4.4 |
| 5 | 0.0007031 | 0.0205083 | 3.4 |
| 2 | 0.0007682 | 0.0083852 | 9.2 |
| 1 | 0.0049353 | 0.0206937 | 23.8 |
| 8 | 0.0076741 | 0.0404725 | 19.0 |
| 6 | 0.0170800 | 0.0772860 | 22.1 |
| 4 | 0.0294116 | 0.0602114 | 48.8 |
| 9 | 0.0343718 | 0.3041610 | 11.3 |
| 7 | 0.0452300 | 0.1587200 | 28.5 |

This is a table of the cross-sectional areas for the SA neck and Spine neck. Surface area is in units of $\mu m^2$.

neck diameters. PSD-to-dendrite distances are constant; at least, for the nine reconstructed spines used in this study.

Finally, we examined the morphological properties of the SAs contained in the reconstructed spines (Fig 2E). We observed a positive correlation between spine volume and SA volume, which was also observed when spine neck cross-sections were considered. These results

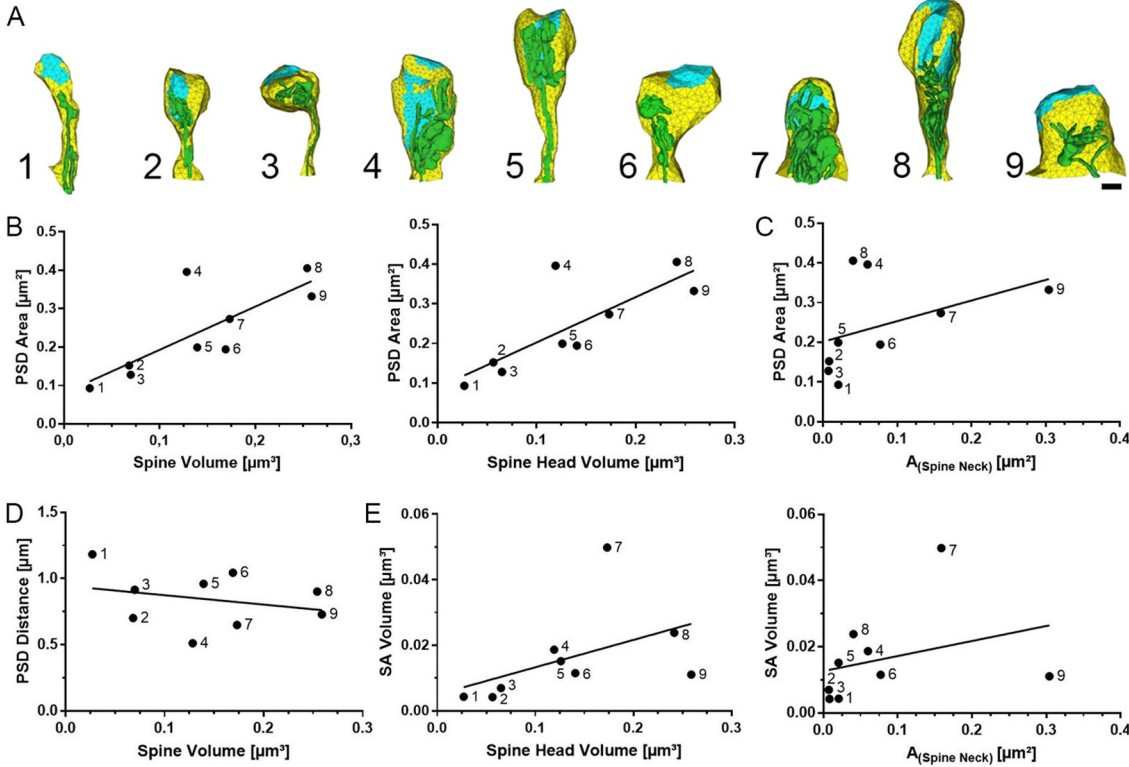

**Fig 2. Morphological parameter of reconstructed spines.** (A) Three-dimensional models based on reconstructed human dendritic spines. Spine apparatus organelle, green. Postsynaptic density, cyan. Scale bar, 200 nm (B) Dendritic spines with large volumes have large postsynaptic densities. (C) Positive correlation between spine neck diameter and postsynaptic density sizes. (D) Postsynaptic densities in large spines are not further away from spine base. (E) Large spines contain large spine apparatus organelles, which are found in spines with wider spine necks. $n$ = 9 reconstructed dendritic spines from 7 independent samples.

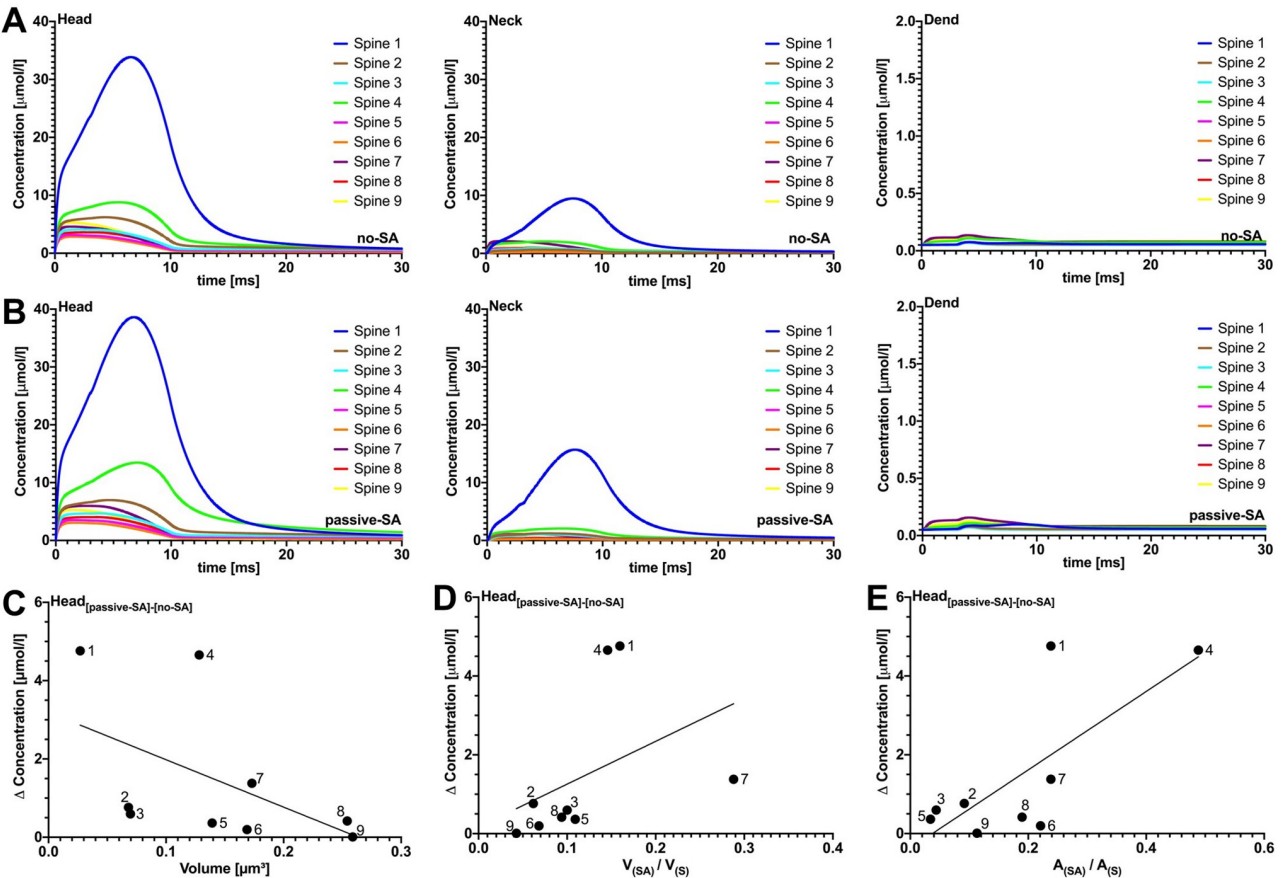

**Fig 3. Passive SA and absent SA.** Effects of passive spine SA on spine-to-dendrite $Ca^{2+}$ signaling, with fixed $Ca^{2+}$ influx. Row A, without SA, and row B, with SA, show $[Ca^{2+}]$ profiles for 10 ms initial $Ca^{2+}$ release into the spine head. Consistent with experimental data, spine-to-dendrite $Ca^{2+}$ signaling does not occur for all 9 simulated spines, and this also holds true for simulations where $Ca^{2+}$ influx is adapted to the postsynaptic area, see additional plots. Row C (fixed $Ca^{2+}$ influx) demonstrates correlations between maximum $[Ca^{2+}]$ in the head region versus spine volume, ratio of SA area/Spine area, and SA volume/Spine volume. Presence of a passive SA, i.e., no ryanodine receptors, has no major effect on $[Ca^{2+}]$ dynamics in the spine head and neck.

provided ultrastructural morphological insight into the three-dimensional organization of SA-containing dendritic spines in the human cortex. They support the previously proposed idea of a spine-within-a-spine ER morphology, which suggests a crucial role for spine volume and sER volume ratio in $Ca^{2+}$ signaling [1, 19].

## Passive spine apparatus organelles have no major impact on spine-to-dendrite $Ca^{2+}$ signaling

To assess the role of the SA in spine-to-dendrite $Ca^{2+}$ communication, we first investigated $Ca^{2+}$ signal propagation in spines for which the reconstructed SA was excluded. In all simulations the same $Ca^{2+}$ ion flux ($j_c^{rls}$) was used for the PSD at the spine head, [6, 33], see Table 1 for the flux value. Changes in $[Ca^{2+}]$ were measured in three regions of interest: the spine head, the spine neck, and the dendrite [20]. For all of our numerical experiments we simulated a glutamate receptor model at the PSD with a 10 ms release of $Ca^{2+}$ by a linearly decreasing $Ca^{2+}$ pulse, described further in the *methods* section.

As shown in Fig 3A, only a small proportion of released $Ca^{2+}$ ions reached the dendritic compartment. To quantify the dendritic response to $Ca^{2+}$ release at the PSD, the $[Ca^{2+}]$

**Table 4. Peak calcium concentrations in measuring zones.**

| Spine | Head | | | Dendrite | | |
|---|---|---|---|---|---|---|
| | passive | active | % Increase | passive | active | % Increase |
| 1 | 33.85 | 44.00 | 30% | 0.08 | 1.64 | 1,950% |
| 2 | 6.22 | 9.66 | 55% | 0.07 | 1.89 | 2,600% |
| 3 | 4.09 | 11.47 | 180% | 0.07 | 1.85 | 2,543% |
| 4 | 8.80 | 24.82 | 180% | 0.11 | 1.06 | 864% |
| 5 | 3.16 | 10.22 | 223% | 0.07 | 1.70 | 2,329% |
| 6 | 2.86 | 6.47 | 126% | 0.07 | 1.62 | 2,214% |
| 7 | 4.60 | 17.09 | 270% | 0.14 | 1.01 | 621% |
| 8 | 3.62 | 8.11 | 124% | 0.07 | 0.88 | 115% |
| 9 | 5.29 | 7.20 | 36% | 0.12 | 1.05 | 775% |

All concentrations are measured in $\mu$mol/$L$.

amplitude was measured in the spine head, neck and dendrite. The values and ratios between head and dendrite amplitudes for all spines are listed in Table 4. In all cases less than 3% of the initial [$Ca^{2+}$] amplitude in the head was detectable in the dendrite. When the SA was added as a passive cellular compartment (i.e., when it was present as a geometric obstacle, but without any $Ca^{2+}$ exchange mechanisms that would allow $Ca^{2+}$ exchange across the SA membrane), similar and near-zero dendritic [$Ca^{2+}$] profiles were observed in the respective spines (Fig 3B). Minor differences in the profiles (compared to the no-SA case) can be observed in the head and neck; however, these all lie in the single-digit $\mu$mol/$l$ range.

The differences in peak [$Ca^{2+}$] amplitude between passive-SA and no-SA are presented in Fig 3C, 3D and 3E. These differences decreased with increasing spine volume (Fig 3C) and were positively correlated with the SA/Spine volume ratio (Fig 3D). Two clear outliers in this trend are spines 1 and 4. To test whether the SA is able to block $Ca^{2+}$ passage at the intersection between head and neck [34], and thus produce larger differences in the [$Ca^{2+}$] amplitude, the data was plotted against the SA/spine surface area ratio at this intersection (Fig 3E). The outlier, spine 4, could clearly be explained by this metric. Thus, accumulation of SA in the intersectional region between head and neck may obstruct $Ca^{2+}$ diffusion. Numerical simulations were also carried out for PSD adjusted $Ca^{2+}$ influx (see Fig 4)- i.e. $Ca^{2+}$ influx was scaled by PSD surface area, using spine 8 as the reference spine. It should be noted that PSD-adjusted $Ca^{2+}$ influx may compensate for the $Ca^{2+}$ accumulation in the head region. In Fig 4, for example, the change in [$Ca^{2+}$] remains relatively uniform when correlated against spine volume, with the exception of spine 4 for which we observe the above-described "corking effect". Taken together, these results (fixed $Ca^{2+}$ and PSD-adjusted), suggest that spine-to-dendrite $Ca^{2+}$ signaling in the dendrite is negligible for both the no-SA and the passive-SA setting, please see the supplemental videos for an animation of the calcium accumulation in the spine region and negligible accumulation in the dendritic region. These observations strengthen the case for active $Ca^{2+}$ exchange across the spine ER membrane in order to enable spine-to-dendrite $Ca^{2+}$ communication in SA-containing dendritic spines. We collected parameters from several previous studies [20, 33, 35–40] to calibrate our model. The parameters varied in this study were the RyR density, maximum calcium influx, and buffering capacity, in order to dissect how different morphological configurations may affect calcium transients and peak calcium amplitudes. In particular, it should be noted that one of our spines (spine 1) produced a maximum calcium amplitude of approximately 50 $\mu$M which is in contrast to previously

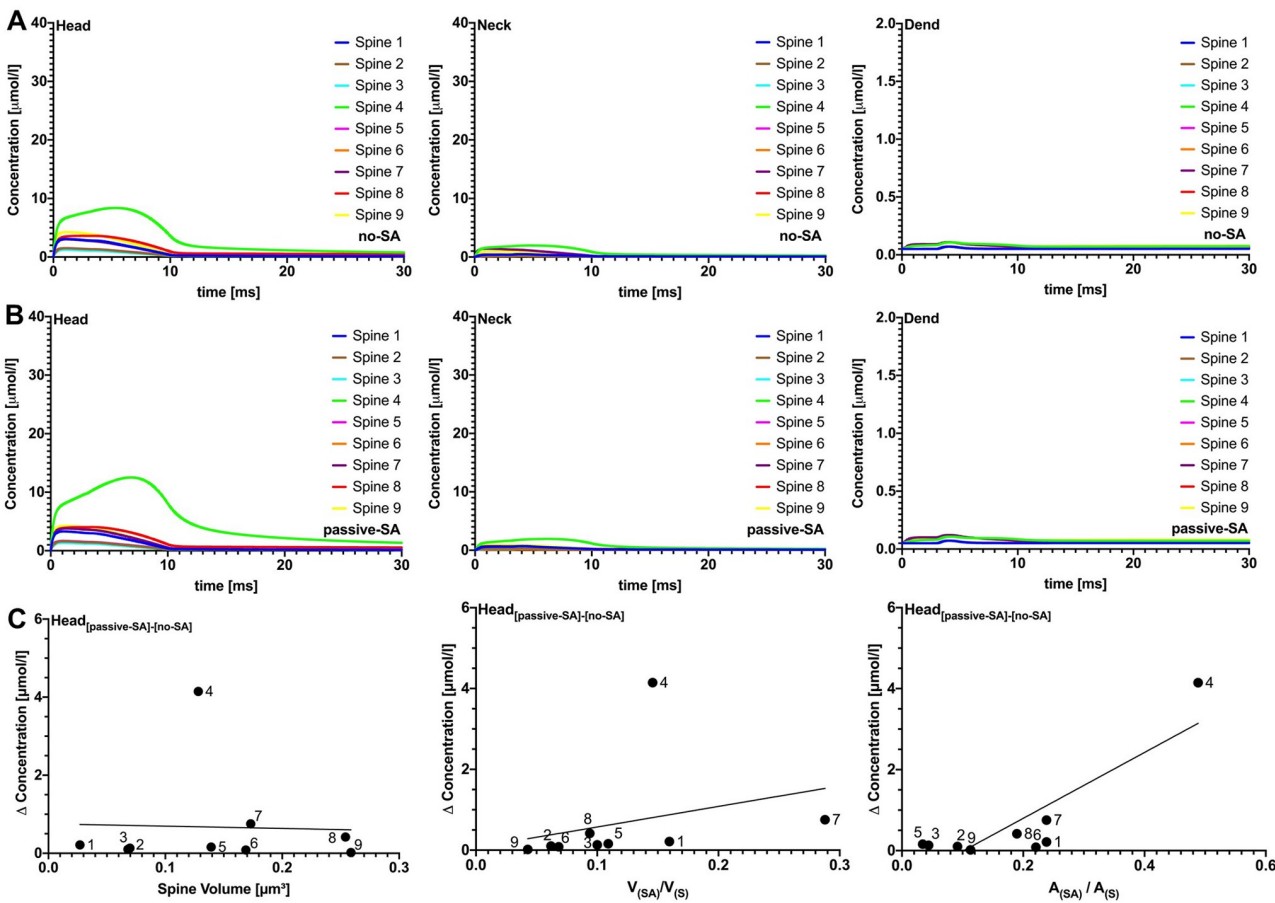

**Fig 4. Passive spine SA spine-to-dendrite signaling.** Effects of passive spine SA on spine-to-dendrite Ca$^{2+}$ signaling, with PSD adjusted Ca$^{2+}$ influx. Plots in row A are without SA and plots in row B are with SA, showing [Ca$^{2+}$] profiles for 10 ms initial Ca$^{2+}$ release into the spine head. Consistent with experimental data, spine-to-dendrite Ca$^{2+}$ signaling does not occur for all 9 simulated spines. Presence of a passive SA, i.e., no ryanodine receptors, has no major effect on Ca$^{2+}$ dynamics in the spine head and neck.

observed values reported in [41]. This discrepancy can be explained by the fact that a constant calcium influx was used for each spine (Fig 3), so that spine 1, with an exceptionally large PSD (the total number of calcium ions released scales with PSD size) together with a small spine head volume, presents a strong outlier (see S2 Fig). After adjusting calcium influx for PSD size the maximum calcium amplitude of all spines was reduced up to 10-fold (see S3 Fig), which confirms previously reported ranges. Given that the [Ca$^{2+}$] in the head scales with the total number of ions released into the spine head, and the size and shape of the spine head, we were interested in spine responses when providing a uniformly calibrated Ca$^{2+}$ signal. We therefore calibrated Ca$^{2+}$ influx for outlier spine 1, such that the peak [Ca$^{2+}$] amplitude was within ranges reported in [41], and used this influx for all spines. As shown in S3 Fig, the peak amplitudes in the head all lay within the reported range, and the Ca$^{2+}$ dynamics in neck and dendrite confirmed the spine to dendrite coupling effects shown in Figs 3 and 4.

## Ryanodine receptors of spine apparatus control spine-to-dendrite Ca$^{2+}$ signaling

Previous work revealed that RyR-dependent Ca$^{2+}$-induced Ca$^{2+}$ release from synaptopodin-associated Ca$^{2+}$ stores modulates spine Ca$^{2+}$ dynamics and synaptic plasticity [7, 12, 13]. To

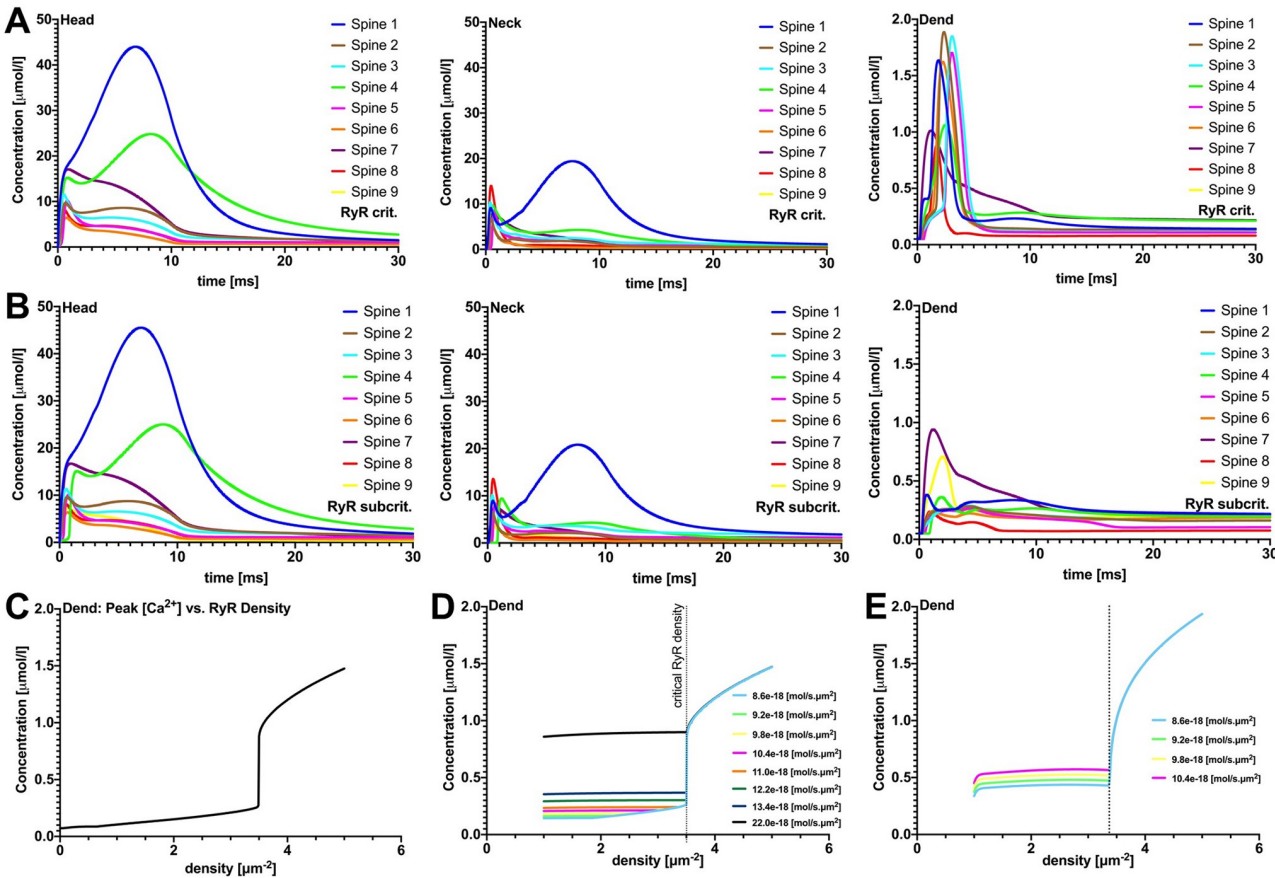

**Fig 5. Effects of active SA with RyR calcium exchange mechanisms only.** These plots were generated with active SA with only RyR. Plots in row A show the calcium concentration traces in the head and dendrite at the critical RyR density. Plots in row B show the calcium concentration traces at subcritical (5% below critical) RyR density. It is worth observing that $Ca^{2+}$ coupling is significantly decreased. In Fig 5C, 5D, and 5E, we demonstrate for spine 8 that there is a critical RyR density such that $Ca^{2+}$ coupling occurs in the dendritic region. The RyR density parameter is incremented in 0.01 $\mu m^{-2}$ steps. The maximum $[Ca^{2+}]$ in the dendritic region has a significant jump at $\approx 3.50\ \mu m^{-2}$, please also see the supplemental video showing an accumulation of calcium in the spine region; however, at critical RyR density calcium is propagated to the dendritic region. For plot D we performed the same RyR density increment experiments but with different $Ca^{2+}$ influx at the synapse. For all spine simulations (rows A and B) we used a fixed $Ca^{2+}$ influx at the synapse.

assess the role of active SAs in human dendritic spines, we included RyRs on the membrane of the SA and dendritic ER, to determine the critical RyR density for spine-to-dendrite $Ca^{2+}$ communication for each spine (Table 2). To quantify spine-to-dendrite $Ca^{2+}$ coupling, we defined the peak $[Ca^{2+}]$ to be a concentration value greater than 1 $\mu mol/L$ in the dendritic measuring zone for 1 ms. Converting this concentration minimum (during 1 ms) leads to $1 \times 10^{-18}$ mol/($s \cdot \mu m^{3}$), and then multiplying by the width of the dendrite ($\sim 1\ \mu m$) gives an approximate flux through the neck-dendritic interface of $1 \times 10^{-18}$ mol/($s \cdot \mu m^{2}$). We defined a spine-to-dendrite coupling as successful if more than 11% of the initial $Ca^{2+}$ influx at the PSD was measured in the dendrite (using $8.6 \times 10^{-18}$ mol/($s \cdot \mu m^{2}$) for reference). We increased the RyR density in 0.01 $\mu m^{-2}$ increments in the range of 0–5 RyRs $\mu m^{-2}$ and found critical RyR values for all reconstructed morphologies, ranging from 1.37 and to 3.5 $\mu m^{-2}$ (Table 2). Fig 5A shows all spines eliciting dendritic $Ca^{2+}$ coupling when the critical RyR density is surpassed. A small number of RyRs on SAs (i.e., 1–6 receptors) was sufficient to enable spine-to-dendrite $Ca^{2+}$ communication, see Table 2. However, the peak $[Ca^{2+}]$ dropped below 1 $\mu mol/L$ when the RyR density is subcritical (Fig 5B). Therefore, synaptic calcium release induces calcium

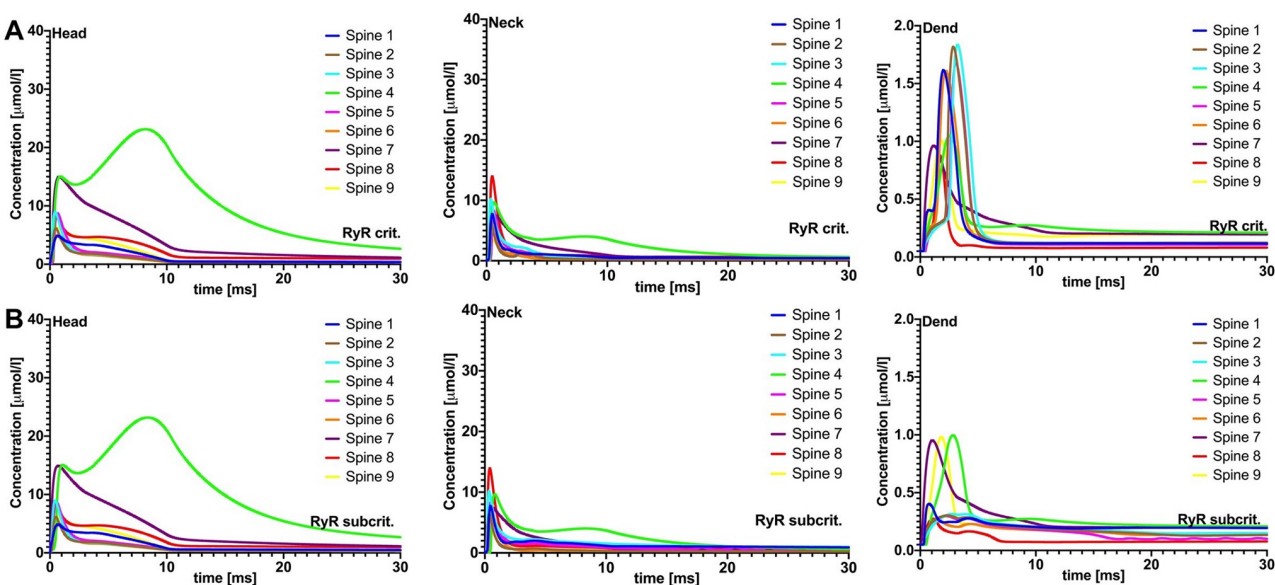

**Fig 6. PSD adjusted calcium influx.** All plots were generated with PSD adjusted calcium influx. In row A, these are the calcium concentration profiles in the head and dendrite region at critical RyR density. In row B, these calcium concentration profiles correspond to sub-critical RyR density.

transients in the dendritic region when there is a critical density of RyRs on the ER membrane. At critical RyR density, depletion of the ER calcium stores occurs due to $Ca^{2+}$-induced $Ca^{2+}$-release. This aligns with observations made previously [8]. Thus, RyR density plays a critical role in spine-to-dendrite $Ca^{2+}$ coupling.

We next tested whether changes in $Ca^{2+}$ entry at the PSD affect the critical RyR density by successively increasing the $Ca^{2+}$ flux density at the PSD entry site. However, coupling occurs at the same RyR density, independent of the initial $Ca^{2+}$ influx within the range tested here ($8.6 \times 10^{-18} - 22.0 \times 10^{-18}$ mol/($s \cdot \mu m^2$), see Fig 5C and 5D for an example). This invariance can be attributed to intracellular $Ca^{2+}$ buffering, as when we ran our simulations with half the buffering capacity (Fig 5E), the critical RyR density was slightly decreased. It is also worth noting that the transition between coupled and decoupled spine-to-dendrite $Ca^{2+}$ signals is highly nonlinear (see jumps in Fig 5C, 5D and 5E). Changes in critical RyR density appear to control an "all-or-nothing" $Ca^{2+}$ dynamics behavior in the parent dendrite.

Comparing the peak calcium concentrations for passive and active SA in the head and dendrites (see Table 4) showed that the relative differences between passive and active ER dynamics are the result of $Ca^{2+}$-induced $Ca^{2+}$ release through RyRs, which releases additional $Ca^{2+}$ into the cytosol [42]. The critical RyR density was not dependent on $Ca^{2+}$ influx at the PSD; we therefore investigated whether the structural organization of dendritic spines could play a role. For completeness, we provided $Ca^{2+}$ profiles for PSD-adjusted $Ca^{2+}$ influx at both critical RyR and subcritical RyR densities (see Fig 6).

## Spine morphology and neck plasticity affect spine-to-dendrite $Ca^{2+}$ signaling

Sufficient $[Ca^{2+}]$ at the endoplasmic membrane is critical for endoplasmic $Ca^{2+}$ release through RyRs [12, 43, 44]. Thus, an intracellular organization of the SA that would support sufficient accumulation of $Ca^{2+}$ at the spine neck (the most confined part of the spine morphology) should reduce the critical RyR density. To test this hypothesis, the critical RyR

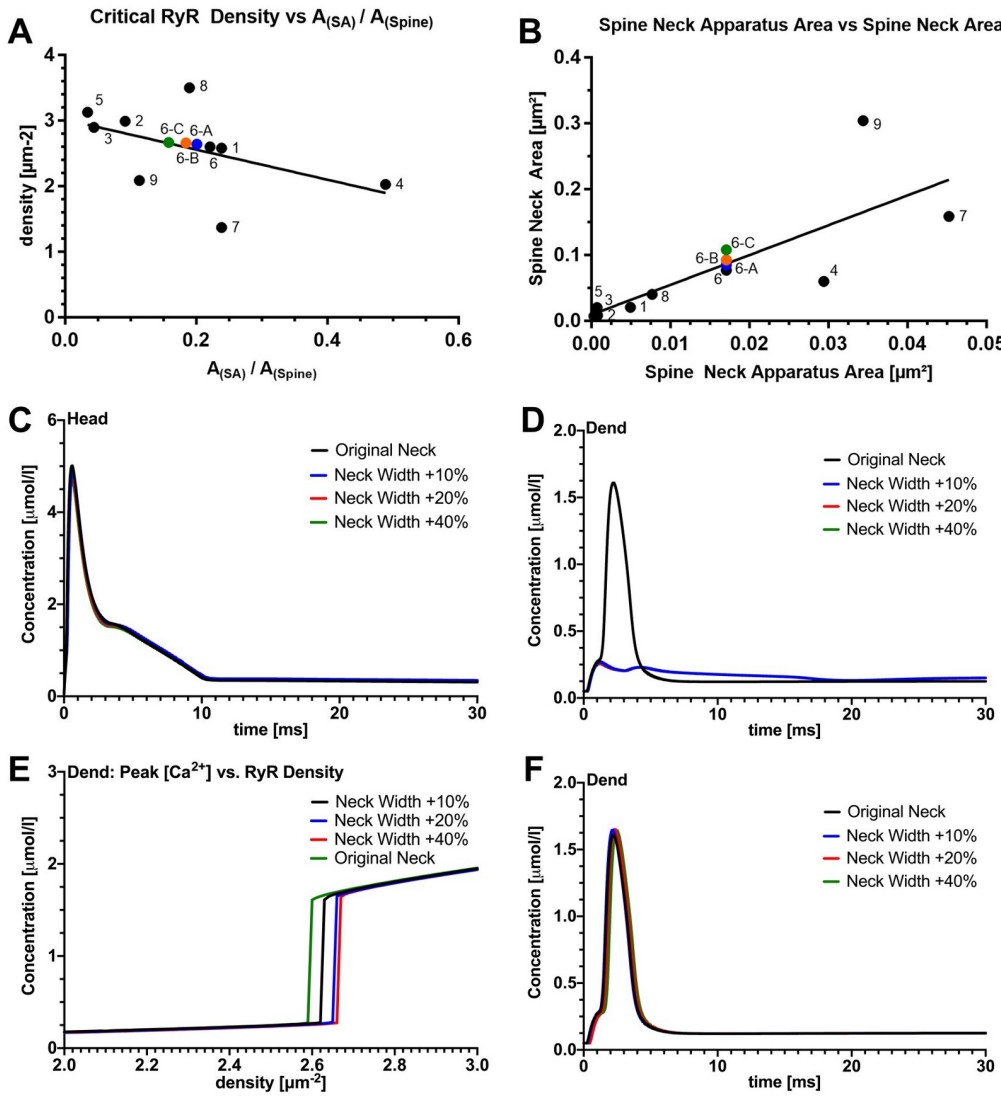

**Fig 7. Critical RyR density.** In A we plot the critical RyR density against the ratio of Spine Apparatus Neck area to Spine Neck area. In Table 2 we show the values that correspond to A. In C and D we demonstrate that widening the neck diameter of a spine affects the coupling of $Ca^{2+}$ to the dendritic region. In particular E indicates that neck widening causes the critical RyR density to increase.

density was plotted against the ratio of SA in the spine neck to the spine neck cross-sectional area (Fig 7A). We found that the area ratio was negatively correlated with the RyR density, demonstrating that a smaller RyR density is sufficient to induce spine-to-dendrite coupling when the SA occupies larger portions of the spine neck.

Interestingly, we observed a positive correlation between the spine and SA cross-sectional area of the spine neck (Fig 7B), suggesting that the SA adjusts to dendritic spine growth and reorganizes in such a way to support spine-to-dendrite $Ca^{2+}$ communication. Previous work has demonstrated that changes in spine neck diameters accompany the induction of excitatory synaptic plasticity, reflected by an increase in spine head size [45–47], and that proximity of the SA to the plasma membrane is a key parameter to SA calcium refilling [8]. Specifically,

spine necks are shortened and widened after long-term potentiation of excitatory neurotransmission [46].

We therefore systematically tested whether changes in spine neck diameter can affect spine-to-dendrite $Ca^{2+}$ communication. To address the role of spine neck plasticity in spine-to-dendrite $Ca^{2+}$ communication, we ran simulations by using the reconstruction of spine 6, and manually widened the spine neck by 10%, 20%, and 40%, while keeping the RyR density and the SA morphology constant (Fig 7, modified spines 6-A, 6-B, 6-C, respectively). While the $Ca^{2+}$ dynamics in the spine head remained unchanged (Fig 7C), we observed that increasing the spine neck width caused a decrease in peak $[Ca^{2+}]$ near the SA membrane, thereby producing insufficient $Ca^{2+}$ release from the SA store to maintain spine-to-dendrite coupling (Fig 7D).

We then tested whether an increase in RyR density could balance morphological changes in order to maintain coupling even under spine growth and reorganization. For each of the modified spines 6-A, 6-B, and 6-C, we incrementally increased the RyR density and observed that a critical RyR density could be determined for all spines. Naturally, this critical value increases with increasing neck width (see Fig 7E; critical RyR densities for spines 6-A, 6-B, 6-C reported in Table 2). We have not tested whether increasing the size of the SA restores spine-to-dendrite communication. These results suggest that for an SA with subcritical RyR density, increasing the SA size will restore spine-to-dendrite communication.

This means, that not only could coupling be restored but, as Fig 7F shows, the dendritic $[Ca^{2+}]$ profiles for all adjusted spines are nearly identical, indicating that $Ca^{2+}$ is sensitive to changes in spine neck and SA morphology in human dendritic spines, and that $Ca^{2+}$ dynamics are affected by spine neck and SA geometry [48–50].

In conclusion, we identified an "all-or-nothing" $Ca^{2+}$ communication switch between spine head and dendrite that is controlled by an interplay between RyR density and the morphology of dendritic spines and SAs.

## Discussion

In this study, we explored spine $Ca^{2+}$ dynamics in human cortical spines using simulations consisting of (i) $Ca^{2+}$ diffusion and reaction with mobile calcium buffer (calbindin) in the cytosol; (ii) $Ca^{2+}$ diffusion inside the SA and sER of the dendrite; and (iii) $Ca^{2+}$ exchange across the plasma (PMCA, NCX) and endoplasmic (RyR) membranes (Fig 1E). We showed that a critical RyR density, dependent on the overall spine and SA architecture, allows active $Ca^{2+}$-induced $Ca^{2+}$ release to maintain spine-to-dendrite $Ca^{2+}$ communication. Increasing the neck diameter or decreasing the SA cross-sectional area in the neck region of individual spines disrupts this communication pathway. Increasing the RyR density (or SA sizes) can reverse this effect (Fig 8). These findings suggest a critical role for SA- and spine neck plasticity in controlling $Ca^{2+}$ communication between the spine head and dendrite.

The ability to visualize individual dendritic spines in living brain tissue over extended periods was developed three decades ago [51, 52]. Consequently, the amount of information about the morphology and physiology of dendritic spines and the molecular mechanisms underlying spine function has increased tremendously [5, 48, 53–55]. However, major issues regarding structure-function interrelations and the role of dendritic spines in network function and complex behavior remain unresolved [55]. This is particularly true for structure-function interrelations in the human cortex, which are beginning to emerge [22]. In adult human neurons, it is currently not possible to carry out the required experiments to study spine $Ca^{2+}$ dynamics as they require genetic tools for simultaneous visualization of dendritic spines, SAs and changes in intracellular $[Ca^{2+}]$ levels. Available data from the human neocortex are mainly based on reconstructions of 2D-projected neurons (e.g., Golgi-stained or intracellularly

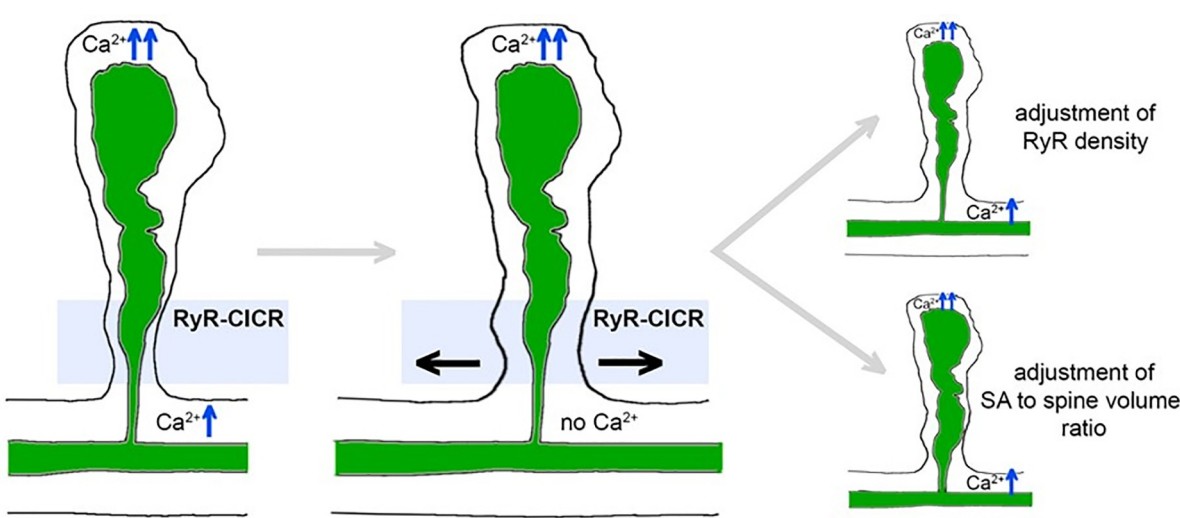

**Fig 8. "All or-nothing" spine-to-dendrite calcium communication.** Schematic illustration summarizing our major findings. Ca²⁺ communication between spine head and dendrite is controlled by the interplay between RyR density, and morphological changes of dendritic spines and spine apparatus organelles (SA) in the neck region. RyR, ryanodine receptors.

injected neurons; [56, 57] and do not capture the complex 3D architecture of SA/ER inside individual spines). Also, it is impossible to systematically assess the relevance of individual spine and SA/ER parameters, as they are not easy to manipulate in biologically complex systems. In this respect, computer simulations based on serial EM reconstructions provide suitable tools to explore spine $Ca^{2+}$ dynamics in human dendritic spines.

Our morphological analyses of the nine reconstructed spines used in this study confirm and expand on our recent findings of dendritic spine plasticity of layer II/III pyramidal neurons in the adult human cortex [22]. In this earlier study, we showed that: (i) approximately 70% of dendritic spines contain synaptopodin clusters, (ii) synaptopodin clusters and spine apparatus organelles are found in dendritic spines with large cross-sectional areas, and (iii) a plasticity-inducing stimulus promotes remodeling of synaptopodin clusters, spine apparatus organelles, and dendritic spines in human cortical neurons. These changes correlated well with changes in spontaneous excitatory postsynaptic currents recorded from individual layer II/III pyramidal neurons [22]. Consistent with these earlier findings, we confirmed in this study that synaptopodin is a marker for the human SA, and we also demonstrated a positive correlation between SA volumes, PSD sizes, and spine volumes. Although this analysis is currently based on only nine reconstructed spines, the ability to detect such interrelations with small numbers indicates that these morphological measures may be tightly regulated in SA-containing spines of the human cortex [58].

Our simulations using a purely passive SA, strengthen the case for active $Ca^{2+}$ exchange across the spine ER membrane to enable spine-to-dendrite $Ca^{2+}$ communication in SA-containing dendritic spines. Specifically, having RyR density above critical value enabled spine-to-dendrite $Ca^{2+}$ communication. This RyR density depended on the overall spine and SA architecture, while being independent of the strength of the initial $Ca^{2+}$ influx through the PSD. Increasing spine neck sizes for individual spines disrupted this communication pathway. Increasing the RyR density, or adjusting the SA to spine-neck ratios, reversed this effect. While the actual RyR densities and changes in the number of RyRs on ER membranes remain unknown, changes in spine neck sizes and spine apparatuses (both sizes and position within a spine) are well described [13, 29, 30, 46, 59]. Hence, the physiology and morphology of

dendritic spines appear tightly integrated and can control $Ca^{2+}$ communication between the spine head and dendrite. In this context, the SA may facilitate $Ca^{2+}$ homeostasis in order to maintain spine-to-dendrite $Ca^{2+}$ communication during dendritic spine growth as seen after plasticity induction. Conversely, it may be that changes in SA size or position within the spine allow for rapid input-specific control of spine $Ca^{2+}$ signals. Whether such mechanisms serve input-specific synaptic plasticity, $Ca^{2+}$ dependent local dendritic protein synthesis [60], synaptic tagging [61, 62], or other as yet unknown mechanisms warrants further investigation. It may be important to mention in this context, however, that in our recent study we showed that blocking protein synthesis with anisomycin halted plasticity induction, and dendritic spine and SA growth in adult human cortical neurons [22], suggesting a role of local protein synthesis in SA-dependent synaptic plasticity.

Interestingly, the transition between coupled and decoupled spine-to-dendrite $Ca^{2+}$ signals was highly nonlinear. This "all-or-nothing" property observed in our simulations is reminiscent of the physiological function of another cellular compartment in which synaptopodin-associated stacked ER is present: the axon initial segment and the cisternal organelle [63–68]. Like SAs, cisternal organelles consist of flat, membrane-delineated cisternae alternating with electron-dense material. Synaptopodin-deficient mice do not form either SAs or cisternal organelles [63]. Indeed, from an anatomical point of view the analogy between spine heads and spine necks, and the soma and axon initial segment (AIS) is striking. However, while the role of SA in spine calcium dynamics and synaptic plasticity seems well-established, the biological significance of cisternal organelles and their role in AIS-plasticity [69–71] remains less well understood. It is likely that increased use of new electron microscopy techniques, such as focused ion beam, block-face serial electron microscopy, and array tomography, together with computer-based automatic 3D reconstructions of human cortical tissue will allow us to advance toward a better understanding of ultrastructure-function relations of synaptopodin-associated stacked smooth ER in distinct neuronal compartments, and to test the extent to which data obtained from animal models translates to the adult human brain.

## Materials and methods

### Ethics statement

Human brain tissue was obtained from a local biobank operated in the Department for Neurosurgery at the Faculty of Medicine, University of Freiburg Germany (AZ 472/15_160880). Written informed consent was obtained from every patient and all procedures were carried out after positive evaluation and approval by the Albert-Ludwigs University of Freiburg ethics committee (AZ 593/19).

### Electron microscopy

Tissue was fixed in 4% paraformaldehyde (w/v) and 2% glutaraldehyde (w/v; phosphate buffered saline) overnight. After fixation tissue was cut into 100 $\mu$m thin slices, thoroughly washed in phosphate buffer (0.1M) and incubated with 1% osmium tetroxide for 60 minutes. After washing in graded ethanol (up to 50% /v/v)) for 10 minutes each, slices were incubated with uranyl acetate (1% (w/v) in 70% (v/v) ethanol) overnight. Slices were then dehydrated in graded ethanol (80%, 90%, 96% for 10 minutes each, and 2x 100% for 10 minutes). After two washing steps in propylene oxide for 5 minutes each, the slices were incubated with durcupan/propylenoxid (1:1 for 1 hour) following durcupan only overnight at room temperature. Ultrathin sections of 20–40 nm were prepared at a Leica UC6 Ultracut. Sections were mounted onto copper grids (Plano) and contrasted using Pb-citrate for 3 minutes. Transmission electron microscopy was performed at a Zeiss Leo 906 equipped at 6000x magnification. In total,

cortical access tissue from 7 individuals who underwent clinically indicated neurosurgical procedures, e.g., for tumors or epilepsy, were processed and 1 to 6 acquisitions were obtained from each patient. Each acquisition included at least one labeled dendritic segment and synaptic contact. Up to 40 images of the same dendritic spines on consecutive sections were acquired and saved as TIF-files.

### 3D-reconstruction of dendritic spines and generation of spine models

Image series were adjusted in brightness and contrast using the Fiji software package [72]. Stack alignment and segmentation were performed with the integrated TrakEM2 plugin [73]. For this purpose, an automatic alignment (rigid registration) was utilized followed by manual correction. Independent traces were drawn manually for each spine attached to a dendritic segment, spine apparatus organelles and postsynaptic densities. Dendritic spines possessing spine apparatus organelles that were complete within the series were reconstructed. A total of 9 series met these criteria and were used in analysis. The image segmentation binary files were transformed into a preliminary surface mesh using a Java 3D library [74] and saved in an 'OBJ' format. However, meshes generated in this manner contained various mesh artifacts, such as jagged boundaries, non-manifold features and surface intersections. Using the meshing software ProMesh [75] for post-processing, including Laplacian smoothing, optimized surface meshes, compatible with physical simulations, were produced. Mesh defects, such as disconnects in the spine apparatus organelle due to thickness in ultrathin section or errors in the segmentation, were manually identified and matched to the cellular structures. Spine structures were manually subdivided into head and neck region after visual inspection of the 3D model from different angles. All reconstructed spines, including their spine apparatus organelles, were cut from their parent dendritic shaft through the base of the spine neck. Subsequently, they were attached to a dendritic segment containing the endoplasmatic reticulum. Using this method, we inserted spine apparatuses into reconstructed human spine models and generated hybrids of anatomical morphologies. To ensure that surface and volume discretizations are suitable for numerical simulations the following procedural steps were followed to process the raw meshes, which consist of the subsets *spine*, *dendrite*, *synapse*, and *ER*. All steps are carried out using [75]:

1. Dendrite surface remeshing with edge lengths in the range 15–30 nm, and the spine, synapse, and ER remeshed with edge lengths within a range of 5–10 nm.

2. Some parts of the ER may intersect the plasma membrane (from the semi-automatic procedure used above with [72, 73] and is corrected by manually moving the vertices/edges of the ER.

3. Consistency checks: remove potential double edges, double vertices, stray vertices, and close holes in the geometry.

4. Remove very small edges (this may arise when remeshing occurs) and mesh holes.

5. Remove sharp surface angles by a) removing critical vertices and remeshing or b) perform one iteration of Laplacian smoothing.

6. Tetrahedral mesh generation with a minimum dihedral angle as large as possible to avoid sliver triangles or obtuse angles.

7. Assign subsets to the geometry for simulation and measuring zones, i.e., cytosol, dendrite, head, etc.

For the Laplacian smoothing we perform the algorithm described in [76], which is the structured Laplacian smoothing algorithm implemented in [75]. Using ProMesh, we measured volumes from the three-dimensional reconstructions. While spine neck length was measured from the center of the spine head base towards the insertion of the spine into the dendrite, neck diameter was calculated as an average diameter from three measurements obtained proximal, intermediate and distal to the edge of the dendrite. Linear regressions were performed by best-fit approaches and were statistically tested to be different from zero with the statistical software GraphPad Prism (GraphPad Software).

## Calcium simulations

All modeling components were implemented in the simulation toolbox NeuroBox [77]. NeuroBox is a simulation toolbox that combines models of electrical and biochemical signaling on one- to three-dimensional computational domains. NeuroBox allows the definition of model equations, typically formulated as ordinary and partial differential equations, of the cellular computational domain and specification of the mathematical discretization methods and solvers [78, 79]. For this work we utilized a continuum-based, deterministic model with a detailed 4-state kinetic RyR model [35], instead of a stochastic modeling approach, with simplified RyR dynamics, as described in previous work found in, e.g., [41]. Using the parameters from [35] we arrive at RyR calcium flux rates of around $3.5 \times 10^{-18}$ mol/$s$ which leads to $\approx 4{,}000$ calcium ions released by all RyRs on the ER membrane. Additionally, previous literature [80–82] highlights the existence of calcium sparks, puffs, and calcium waves; while sparks and puffs are controlled by, potentially stochastic, clustered channel activity our studies focus on the macroscopic effects of calcium wave initiation along the endoplasmic membrane which constitutes a significantly higher number of calcium ions released into the cytosol. The synaptic calcium influx modeling choice is motivated by the use of AMPA receptors that release $\approx 2{,}400$ calcium ions at the PSD (using a reference spine and influx rate of $\approx 10^{-17}$ mol $\cdot s^{-1} \mu m^{-2}$ and a 10 ms time constant). These numbers merit a continuum-based model. Carrying out thousands of Monte Carlo simulations with very high particle numbers and then averaging produces unreasonable computational cost compared to performing the averaging in the model equations first and then computing a continuum-based equation. This is not to say stochastic effects are negligible or of no scientific significance. In fact, non-homogeneously distributed RyR or IP3 receptors and low-calcium concentration events (at the level of single particle simulations) coupled to a continuum model should be considered in forthcoming research but remain outside the scope of this structure-function study. The microscopy images were processed to generate 3D surface mesh geometries. With these surface mesh geometries, we used Promesh [75] to perform a tetrahedralization of the surface geometries [27, 83], in particular we performed a Delaunay tetrahedrization which avoids tetrahedra with particularly acute or obtuse interior angles.

## Model equations

Spatio-temporal calcium dynamics are modeled by a system of diffusion-reaction equations

$$\frac{\partial c_c}{\partial t} = D_c \Delta c_c + (\kappa_b^- (b^{tot} - b) - \kappa_b^+ b c_c), \tag{1}$$

$$\frac{\partial b}{\partial t} = D_b \Delta b + (\kappa_b^- (b^{tot} - b) - \kappa_b^+ b c_c), \tag{2}$$

$$\frac{\partial c_e}{\partial t} = D_c \Delta c_e, \tag{3}$$

where $(c_c)$ denotes $[\text{Ca}^{2+}]$ in the cytosol, $(c_e)$ the concentration in the ER, and $b$ the *calbindin-D$_{28k}$* concentration. The diffusion constants $D$ are defined using data from [84, 85]. The interaction between cytosolic calcium and calbindin-$D_{28k}$ (CalB) are described by

$$\text{Ca}^{2+} + \text{CalB} \overset{\kappa_b^-}{\underset{\kappa_b^+}{\rightleftharpoons}} \text{CalBCa}^{2+} \tag{4}$$

where the concentration of the CalB-Ca$^{2+}$ compound is expressed by the difference of the total concentration of CalB present in the cytosol ($b^{tot}$) and free CalB, the former of which is assumed to be constant in space and time (this amounts to the assumption that free calcium and CalB have the same diffusive properties). All constants and parameters are borrowed from [20] but are provided in Table 1 for completion. While CalB has four distinct high-affinity Ca$^{2+}$-binding sites [86], we treat it as though it had only one, at the same time quadrupling its concentration in our model. This amounts to assuming that all four binding sites are essentially equal and binding is non-cooperative. The rate constants $\kappa_b^+$ and $\kappa_b^-$ are defined in [20].

## Membrane transport mechanisms

In order to study the influence of the intracellular organization on Ca$^{2+}$ signals, we included Ca$^{2+}$ exchange mechanisms on the plasma membrane (PM). For the plasma membrane, we considered plasma membrane Ca$^{2+}$-ATPase pumps (PMCA), Na$^+$/Ca$^{2+}$ exchangers (NCX), calcium release due to voltage-dependent calcium channels (VDCC), and a leakage term. This amounts to the flux equations (number of ions per membrane area and time)

$$j_{ERM} = j_R + j_{l,e}, \tag{5}$$

$$j_{PM} = -j_P - j_N + j_{l,p} \tag{6}$$

where $j_R$ is the RyR flux density and $j_{l,e}$ is the leakage flux density on the ERM, and $j_P$, $j_N$ and $j_{l,p}$ the flux densities of PMCA, NCX, and leakage flux density of the PM, respectively. Homogeneous distributions of all exchange mechanisms were assumed, as experimental data on precise numbers and spatio-temporal distribution of these receptors within individual spines are not available.

**RyR channels.** The focus of this study is to determine the conditions under which stable calcium waves [80] can propagate from spine head into the dendritic shaft. This stands in contrast to the strongly localized calcium events, such as calcium blips, sparks, and puffs, that have been studied in prior work [81, 82]. Ryanodine receptors [35] thus are assumed homogeneously distributed across the ER membrane, leading to roughly 4,000 calcium ions released through RyR during a successful calcium wave event. The Ca$^{2+}$ flux density generated by ryanodine receptor channels at the ER membrane is given by an expression of the form

$$j_R = \rho_R \cdot p_R^o \cdot I_R, \tag{7}$$

where $\rho_R$ is the density of RyR in the ER membrane, $p_R^o$ is the open state probability of a single channel, and $I_R$ is the single channel Ca$^{2+}$ current. Using [35] we defined a single channel as

$$I_R = I_R^{ref} \frac{c_e - c_c}{c_e^{ref}}, \tag{8}$$

where the reference current $I_R^{ref}$ is approximated in [87]. The open probability of the RyR channels is borrowed from [35] and is computed by the sum of two open probabilities $o_1$ and $o_2$

which are governed by the system

$$o_1 = 1 - c_1 - o_2 - c_2 \tag{9}$$

$$\frac{\partial c_1}{\partial t} = k_a^- o_1 - k_a^+ c_c^4 c_1 \tag{10}$$

$$\frac{\partial o_2}{\partial t} = k_b^+ c_c^3 o_1 - k_b^- o_2 \tag{11}$$

$$\frac{\partial c_2}{\partial t} = k_c^+ o_1 - k_c^- c_2 \tag{12}$$

where the kinetic constants $k_a^\pm, k_b^\pm, k_c^\pm$ are given in Table 1 and this system was solved independently on every point of the ER surface geometry.

**PMCA pumps.**   We used the model given in [36, 88], where the plasma membrane $Ca^{2+}$-ATPase current is modeled as a second-order Hill equation

$$j_P = \rho_P \cdot \frac{I_P c_c^2}{K_P^2 + c_c^2}. \tag{13}$$

**NCX pumps.**   For the $Na^+/Ca^{2+}$ exchange current we assumed a constant $Na^+$ concentration at the plasma membrane utilizing a first-order Hill equation from [36, 88]:

$$j_N = \rho_N \cdot \frac{I_N c_c}{K_N + c_c}. \tag{14}$$

**Leakage term.**   The ER membrane and plasma membrane have a leakage flux that needs to be accounted for and incorporated with described transport mechanisms. The leakage fluxes are calculated to ensure there is zero membrane net flux in the equilibrium state for all simulated ions and agents. Leakage flux densities were modeled by

$$\begin{aligned} j_{l,e} &= v_{l,e} \cdot (c_e - c_c), \\ j_{l,p} &= v_{l,p} \cdot (c_o - c_c), \end{aligned}$$

where $c_o$ is the extracellular $Ca^{2+}$ concentration, which is assumed constant.

**Calcium influx.**   The $Ca^{2+}$ influx is modeled using Neumann boundary conditions for the cytosolic $Ca^{2+}$ concentration, i.e. the time-dependent influx density function is defined at the postsynaptic portion of the plasma membrane, an AMPA model. For our experiments we modeled release of $Ca^{2+}$ by a linearly decreasing $Ca^{2+}$ pulse for $\tau_{rls} = 10$ millisecond duration starting at an initial maximal specific current density $j_c^{rls}$

$$j_c^{rls} \cdot \left(1 - \frac{t}{\tau_{rls}}\right). \tag{15}$$

We assumed a constant membrane potential $V_m = -70$ mV.

## Numerical methods

For numerical simulations, the four equations were discretized in space using a finite volume method. Current densities, both synaptic and across the ER and plasma membranes, were

incorporated into the reaction-diffusion process very naturally and easily this way. We show how this is achieved using the cytosolic [Ca$^{2+}$] equation as an example: It is reformulated (using the divergence theorem) to an integral version

$$\int_B \frac{\partial c_c}{\partial t} \; dV = \int_{\partial B} D_c \nabla^T c_c \cdot n_{\partial B} \; dS + \int_B \left( \kappa_b^-(b^{tot} - b) - \kappa_b^+ b c_c \right) \; dV \tag{16}$$

where $B$ is a control volume that will be specified shortly, and $n_{\partial B}$ is the outward normal on the boundary of $B$. For control volumes located at the ER membrane, some portion of its boundary will coincide with the ER membrane. Since there is no diffusive flux density $D_c \nabla c_c$ across the ER membrane, we could simply substitute it by the ER flux density $j_{ERM}$ into the boundary integral for this portion of the boundary. The same applies to the plasma membrane and the synapse area. The diffusive flux is set to zero on the rest of the cytosolic domain boundary. If we denote the cytosolic boundary by $\Gamma$, its ER/plasma membrane and synaptic parts by $\Gamma_{ERM}$, $\Gamma_{PM}$ and $\Gamma_{syn}$, respectively, this yields the following equation:

$$
\begin{aligned}
\int_B \frac{\partial c_c}{\partial t} \; dV \;\; = & \int_{\partial B \setminus \Gamma} D_c \nabla^T c_c \cdot n_{\partial B} \; dS + \int_{\partial B \cap \Gamma_{ERM}} j_{ERM}^T \cdot n_{\partial B} \; dS \\
& + \int_{\partial B \cap \Gamma_{PM}} j_{PM}^T \cdot n_{\partial B} \; dS + \int_{\partial B \cap \Gamma_{syn}} j_{syn}^T \cdot n_{\partial B} \; dS \\
& + \int_B \left( \kappa_b^-(b^{tot} - b) - \kappa_b^+ b c_c \right) \; dV
\end{aligned}
\tag{17}
$$

Control volumes were constructed as a Voronoi-like dual tesselation of the original tetrahedral mesh by connecting the mid-points of edges, faces and volumes through planar facets. Eq (17) must hold for all control volumes, giving rise to one equation per control volume. Time discretization was realized using a backwards Euler scheme, i.e., for each point in time $t$, the term $\frac{\partial c_c}{\partial t}$ in (17) is replaced by the discretized term $\frac{c_c(t) - c_c(t-\tau)}{\tau}$ and all quantities on the right-hand side were evaluated at time $t$. Here, $\tau$ is the time-step size of the time discretization. By limiting the function space to the space of continuous functions that are linear on all volumes of the original mesh, the integrals in Eq (17) can be evaluated efficiently. Moreover, the solution can be represented by one degree of freedom per volume, so there is one equation for each degree of freedom. The system of equations arising from this procedure is nonlinear (due to the nonlinear reaction term and, more importantly, the highly nonlinear transport terms across the membranes) and is therefore linearized by a Newton iteration. For the presented results, the linearized problems were solved using a Bi-CGSTAB [89] solver using Gauss-Seidel preconditioning, while time integration was realized using the implicit backward Euler scheme. The Bi-CGSTAB iterative solve was set to a maximum of 10,000 iterations and is used to solve the implicit problem at every time step, i.e. Bi-CGSTAB was utilized for an inner solve. The implicit problem can be written as an implicit update rule

$$c_c^{(t)} = c_c^{(t-\tau)} + \tau F(c_c^{(t)})$$

where $t$ is the next time step, $t - \tau$ is the current time step, and $F(\cdot)$ is the right-hand side of the system of equations. For the time integration we used adaptive time-stepping with a prescribed simulation time step size $\tau_{sim}$, where a change in the time step size is governed by performing a convergence check for the implicit solve of the nonlinear system of equations. For each simulation the minimum time step size is set by $\tau_{min} = \tau_{sim}/2^{15}$. The Euclidean norm of the residual

was used to check for convergence. The implicit equation is of the form

$$
\begin{aligned}
c_c^{(t)} - \tau F(c_c^{(t)}) - c_c^{(t-\tau)} \quad &\approx c_c^{(t)} - \tau M c_c^{(t)} - c_c^{(t-\tau)} \\
&= (I - \tau M)c_c^{(t)} - c_c^{(t-\tau)} = 0 \\
&\Rightarrow Ax - b = 0
\end{aligned}
$$

where $r_n = Ax_n - b$ is the residual [90], $x_n$ is the approximation after $n$ iterations of Bi-CGSTAB, using a tolerance of $\epsilon = 10^{-18}$. This tolerance was chosen to ensure accuracy of the linearized problem and to ensure convergence of the trajectories. In the equation above the matrix $M$ is the matrix corresponding to the linearized problem of $F(\cdot)$ using a Jacobi matrix. When we collect all the terms to one side of the equality we obtain an equation of the form $Ax - b = 0$. We performed convergence analysis by a sequence of test simulations in which the time step size $\tau$ was halved (S1 Fig). In S1 Fig we show the convergence of the solutions by decreasing the time step size by powers of 2 (see S1 Fig) and decreasing the error tolerance in S1(C) Fig. In S1 Fig we choose the smallest tolerance such that the difference in solutions were within machine precision. Based on the convergence results we chose a standard time step size of $\tau = 0.5\ \mu s$ since differences in the calcium concentrations between this value and the next were less than 1/100th $\mu M$ and there was no discernible difference with our determined critical ryanodine receptor values. Computations were facilitated by a domain decomposition parallelization approach and carried out using the UG4 framework [79].

## Compute resources

This research includes calculations carried out on HPC resources supported in part by the National Science Foundation through major research instrumentation grant number 1625061 and by the US Army Research Laboratory under contract number W911NF-16–2-0189. The numerical simulations for this work were also carried out using the Extreme Science and Engineering Discovery Environment (XSEDE) see [91], which is supported by National Science Foundation grant number ACI-1548562. Through XSEDE we used the SDSC Comet HPC resources, allocation identification DMS200031.

## Supporting information

**S1 Video. Inactive spine apparatus.** In this video we demonstrate with inactive ER and SA, there is minimal calcium propagation to the dendritic region.
(MP4)

**S2 Video. Active spine apparatus.** In this video we demonstrate with active ER and SA, and super critical RyR density there is calcium propagation to the dendritic region.
(MP4)

**S1 Fig. Convergence plots.** (A) Convergence of the numerical solution using the non-refined geometry, (B) convergence of the numerical solution using a refined geometry, (C) convergence using progressively smaller residual error tolerance. For (A) and (B) we decreased $\Delta t$, the maximum time step size, by powers of 2. We determined convergence when the difference in numerical solutions were near or within machine precision.
(TIF)

**S2 Fig. Concentration versus PSD area/Head volume ratio.** Peak calcium concentrations increase linearly with the ratio of PSD area to spine head volume. A clear outlier is spine 1 which produced above average peak calcium amplitudes in the spine head compared to all

other spines. This is due to the fixed calcium influx with a particularly large PSD area, but small head volume.
(TIF)

**S3 Fig. Calcium dynamics using calibrated calcium influx.** We studied the calcium responses in head, neck, and dendrite of all spines under calibrated calcium entry profiles in order to confirm that calcium profiles previously reported in [41] support our findings. For this we calibrated spine 1 to produce previously reported peak amplitudes between $0 - 14 \, \mu$M in the head and used this setting on all spines. (A) Calcium dynamics in the spine head, neck, and dendrite, when no ER is present in the spine, (B) when a passive ER is added, and (C) when an active ER at critical RyR density is added. The results confirm the reported spine to dendrite coupling dynamics.
(TIF)

## Acknowledgments

We thank Barbara Joch and Sigrun Nestel for excellent technical assistance in electron microscopy.

## Author Contributions

**Conceptualization:** Gillian Queisser, Andreas Vlachos.

**Data curation:** James Rosado, Viet Duc Bui.

**Funding acquisition:** Gillian Queisser, Andreas Vlachos.

**Investigation:** James Rosado, Viet Duc Bui, Gillian Queisser.

**Methodology:** James Rosado, Viet Duc Bui.

**Project administration:** Gillian Queisser, Andreas Vlachos.

**Resources:** Carola A. Haas, Jürgen Beck, Gillian Queisser, Andreas Vlachos.

**Software:** James Rosado.

**Supervision:** Gillian Queisser.

**Visualization:** James Rosado, Viet Duc Bui.

**Writing – original draft:** James Rosado, Viet Duc Bui, Gillian Queisser, Andreas Vlachos.

**Writing – review & editing:** James Rosado, Viet Duc Bui, Carola A. Haas, Jürgen Beck, Gillian Queisser, Andreas Vlachos.

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
