## [Decision Letter · Decision Letter 0]

17 Nov 2021

Dear Mr. Rosado,

Thank you very much for submitting your manuscript "Calcium modeling of spine apparatus-containing human dendritic spines demonstrates "all-or-nothing" communication switch between the spine head and dendrite" for consideration at PLOS Computational Biology.

As with all papers reviewed by the journal, your manuscript was reviewed by members of the editorial board and by several independent reviewers. In light of the reviews (below this email), we would like to invite the resubmission of a significantly-revised version that takes into account the reviewers' comments. Specifically you need to address remarks of reviewers 1 and 2 with additional simulations and data analysis if necessary.

We cannot make any decision about publication until we have seen the revised manuscript and your response to the reviewers' comments. Your revised manuscript is also likely to be sent to reviewers for further evaluation.

Sincerely,

Joanna Jędrzejewska-Szmek, Ph.D.

Associate Editor

PLOS Computational Biology

Kim Blackwell

Deputy Editor

PLOS Computational Biology

Reviewer's Responses to Questions

**Comments to the Authors:**

Reviewer #1: In the current study the authors use EM-based reconstructions of 9 human dendritic spines containing spine apparatuse organelles to model calcium dynamics in dendritic spine. The anatomical study of reconstructed human dendritic spines revealed that the size of the postsynaptic density correlates with spine head volume and that the spine apparatus volume increases alongside the spine volume. Next, they linked these findings to spine-to-dendrite calcium communication. They showed that the calcium-induced calcium release from this intracellular organelle allows for finely tuned "all-or nothing" spine-to-dendrite calcium coupling and this process is controlled by spine morphology, neck plasticity, and ryanodine receptors.

Being a neuroscientist I am not able to assess the accuracy of the modeling part of the study. However the motivation of the study and biological material used rise several questions which should be addresses prior to the publication.

- Why do authors focus on human dendritic spines? Are there any significant differences in morphology and/or other properties between human and rodent dendritic spines that could affect the calculations? Do authors expect or know that calcium dynamics in rodent dendritic spines is different? These problems should be discussed.

- It remain unclear why the authors use only 9 human dendritic spines while the published data describing morphology of rodent dendritic spines is much richer (e.g. PMID: 34428203). If the authors chose only 9 dendritic spines how they can be sure that such spines represent “the common types” in the population, and therefore the conclusions drown from the model are correct and can be generalized.

- Accordingly, the conclusions about the tight correlations between spines parameters can be wrong due to low n number. This has been recently shown for the rodent dendritic spines (PMID: 30737431).

Reviewer #2: In this manuscript, the authors take on a very large task of reconstructing dendritic spines from human tissue samples, reconstructing spine apparatuses from these samples, and using these geometries in simulations. Because the main conclusions rely entirely on simulations and rightly so, as the authors point out that experiments cannot be conducted in these cases, it is then important to recognize that the accuracy of the results and their interpretations rely heavily on the computational schemes used. This is where it is not clear to this reviewer if this was the best approach.

Major points:

1) The authors claim that spine apparatus sets up an ‘all-or-none’ connection between the spine head and the dendrite. But others have shown that only 14-19% of spines have a spine apparatus in other species(Kristen Harris’ work) and I’m not sure this is even known for humans. The fraction of spines surveyed that had a spine apparatus was not mentioned in the text. But from the conclusion, it would seem that those spines without spine apparatuses are at a disadvantage in humans. Is this the suggestion that the authors are making?

2) The authors use a deterministic model for calcium dynamics but it is well known that in small volumes, the deterministic limit may not apply. Particularly so in the SA. The authors don’t mention the impact of their model choice particularly since this was the focus of extensive study — https://www.frontiersin.org/articles/10.3389/fnsyn.2015.00017/full

3) Similarly RyR dynamics are known to be stochastic (see for example https://rupress.org/jgp/article/153/4/e202012685/211900/In-silico-simulations-reveal-that-RYR-distribution) and I found it quite surprising that there was no mention of this.

4) Calibration with known calcium dynamics from the literature — this to me was the strangest part of the model — the authors show peaks of up to 50 \\mu M!! The model was not calibrated to known calcium transients in the literature nor where any clarifications provided about how the different parameters would affect the outcome.

5) Mesh generation: How do the mesh generation methods preserve the original geometry? This is not discussed. Generating surface meshes is actually not as easy as the authors make it out to be — see https://journals.plos.org/ploscompbiol/article?id=10.1371/journal.pcbi.1007756 For example.

6) Numerical convergence — how do the authors know that the simulations converged despite all the geometric complexity and non linearity of the equations?

Reviewer #3: The paper presents a three dimensional computational model of human dendritic spines.

The model is used for investigating human spine and dendrite computationally. A diffusion-reaction model is used in order to carry out different parameter studies for the calcium dynamics in spine and dendrite. The results are reported.

The paper is well written and clearly understandable and presents interesting parameter studies.

The functioning of spines and dendrites is often modeled using ODEs, i.e. with simple models which do not take geometric aspects into account. In this paper, the authors base their findings on partial differential equations (PDEs), which requires significantly more numerical competences and physiological knowledge.

A well balanced pipeline from imaging over modeling to HPC simulation is necessary in order to achieve the presented results. To my knowledge, there is not much literature around which is combining state-of-the arte numerics for PDEs and reaction models for the investigation of spines and dendrites as is done in this paper. I therefore recommend to publish this paper.

Typo:

3D-reconstrunction -> 3D-reconstruction

**Have the authors made all data and (if applicable) computational code underlying the findings in their manuscript fully available?**

Reviewer #1: Yes

Reviewer #2: Yes

Reviewer #3: None

PLOS authors have the option to publish the peer review history of their article (what does this mean?). If published, this will include your full peer review and any attached files.

Reviewer #1: No

Reviewer #2: No

Reviewer #3: No
---

## [Decision Letter · Decision Letter 1]

8 Feb 2022

Dear Mr. Rosado,

Thank you very much for submitting your manuscript "Calcium modeling of spine apparatus-containing human dendritic spines demonstrates an "all-or-nothing" communication switch between the spine head and dendrite" for consideration at PLOS Computational Biology.

As with all papers reviewed by the journal, your manuscript was reviewed by members of the editorial board and by several independent reviewers. In light of the reviews (below this email), we would like to invite the resubmission of a significantly-revised version that takes into account the reviewers' comments. The authors need to address all the reviewer 2 comments:

incorporate in the manuscript the authors' response to comment 2 from the first revision (the reader has to be sure, why the deterministic approach to simulating calcium in the spine is not incorrect),address comment 4 with additional simulations showing that the model is well calibrated and explain 50 uM Ca in the spine in Fig. 5,possibly add some stochastic simulations to further assure the reader that stochastic effects do not play a huge role in their simulations.

We cannot make any decision about publication until we have seen the revised manuscript and your response to the reviewers' comments. Your revised manuscript is also likely to be sent to reviewers for further evaluation.

Sincerely,

Joanna Jędrzejewska-Szmek, Ph.D.

Associate Editor

PLOS Computational Biology

Kim Blackwell

Deputy Editor

PLOS Computational Biology

Reviewer's Responses to Questions

**Comments to the Authors:**

Reviewer #1: The authors addressed all my questions in a satisfactory way.

Reviewer #2: I am honestly not sure what to make of the revised manuscript. In the response to reviewers, the authors agree that all the concerns I raised are valid and important. But the revised manuscript has minimal text changes but a very minimal attempt to ponder if considerations of the concerns raised could alter the results and predictions. I’ll provide a specific example — the authors state that they choose not to calibrate parameters to ensure that the calcium concentrations were constrained to prior observations. “ We intentionally did not tune the other model parameters to fit experimental transients observed under different conditions, since we were interested in studying how different morphological configurations may affect calcium transients and peak amplitudes. ” This is worrisome because the strength of the conclusions and predictions rely on such calibrations. How can the role of morphological configurations be assessed if the model is not calibrated?

In my assessment, it appears that in the private response to reviewers, the authors agree that all the points raised, in fact, by both reviewers are quite important. But in the public document, which is the manuscript, the concerns appear to be swatted away.

Reviewer #3: The revised version extends the papers in terms of properties of the model and its application. As a numerical reviewer, I find the added discussion points very helpful in order to understand better the model choices made, but I cannot comment on the biological aspects.

The question concerning the possible use of stochastic models are well justified and reasonably answered

by the authors from my point of view.

I also appreciate the added section on the numerical aspects of the solution process. I suggest nevertheless to clarify some details (see below) in a minor revision.

Remarks:

- The measure of the residual norm was used to determine the value of the convergence check.

Please reformulate. Maybe: "The Euclidean norm of the residual was used to check for convergence"?

10e-18 is a quite strict tolerance. It is "nice to have", but is it necessary? Is this high accuracy for the linearized problems required in order to get convergence of the resulting trajectories?

- The implicit equation is of the form

c_c(t-\\tau)+\\tau F(c_c(t),t) = 0 => Ax-b =0

where r n= Ax n − b is the residual

Please clarify the relation between A and F. I assume that A is the linearized Model (Jacobi matrix)? A Newton iteration is applied in each time step with bi-cgstab as inner solver? Or is only one linear problem solved per time step?

**Have the authors made all data and (if applicable) computational code underlying the findings in their manuscript fully available?**

Reviewer #1: Yes

Reviewer #2: Yes

Reviewer #3: None

PLOS authors have the option to publish the peer review history of their article (what does this mean?). If published, this will include your full peer review and any attached files.

Reviewer #1: No

Reviewer #2: No

Reviewer #3: No
---

## [Editor Report · Decision Letter 2]

30 Mar 2022

Dear Mr. Rosado,

We are pleased to inform you that your manuscript 'Calcium modeling of spine apparatus-containing human dendritic spines demonstrates an "all-or-nothing" communication switch between the spine head and dendrite' has been provisionally accepted for publication in PLOS Computational Biology.

Best regards,

Joanna Jędrzejewska-Szmek, Ph.D.

Associate Editor

PLOS Computational Biology

Kim Blackwell

Deputy Editor

PLOS Computational Biology

---

## [Editor Report · Acceptance letter]

21 Apr 2022

PCOMPBIOL-D-21-01839R2 

Calcium modeling of spine apparatus-containing human dendritic spines demonstrates an "all-or-nothing" communication switch between the spine head and dendrite

Dear Dr Rosado,

I am pleased to inform you that your manuscript has been formally accepted for publication in PLOS Computational Biology. Your manuscript is now with our production department and you will be notified of the publication date in due course.

With kind regards,

Livia Horvath
